# Walnut Oil Prevents Scopolamine-Induced Memory Dysfunction in a Mouse Model

**DOI:** 10.3390/molecules25071630

**Published:** 2020-04-02

**Authors:** Jianqiao Liao, Yifan Nai, Li Feng, Yimeng Chen, Mei Li, Huaide Xu

**Affiliations:** College of Food Science and Engineering, Northwest A&F University, Yangling 712100, China; ljq1150163714@163.com (J.L.); 2014014831@nwsuaf.edu.cn (Y.N.); fengli0304@163.com (L.F.); cym18049222782@163.com (Y.C.)

**Keywords:** walnut oil, scopolamine-induced, memory impairment, cholinergic system, oxidative stress

## Abstract

For thousands of years, it has been widely believed that walnut is a kind of nut that has benefits for the human body. Walnut oil, accounting for about 70% of walnut, mainly consists of polyunsaturated fatty acids. To investigate the effect of walnut oil on memory impairment in mice, scopolamine (3 mg/kg body weight/d) was used to establish the animal model during Morris Water Maze (MWM) tests. Walnut oil was administrated orally at 10 mL/kg body weight/d for 8 consecutive weeks. The results showed that walnut oil treatment ameliorated the behavior of the memory-impaired mice in the MWM test. Additionally, walnut oil obviously inhibited acetylcholinesterase activity (1.26 ± 0.12 U/mg prot) (*p* = 0.013) and increased choline acetyltransferase activity (129.75 ± 6.76 U/mg tissue wet weight) in the brains of scopolamine-treated mice (*p* = 0.024), suggesting that walnut oil could prevent cholinergic function damage in mice brains. Furthermore, walnut oil remarkably prevented the decrease in total superoxide dismutase activity (93.30 ± 5.50 U/mg prot) (*p* = 0.006) and glutathione content (110.45 ± 17.70 mg/g prot) (*p* = 0.047) and the increase of malondialdehyde content (13.79 ± 0.96 nmol/mg prot) (*p* = 0.001) in the brain of scopolamine-treated mice, indicating that walnut oil could inhibit oxidative stress in the brain of mice. Furthermore, walnut oil prevented histological changes of neurons in hippocampal CA1 and CA3 regions induced by scopolamine. These findings indicate that walnut oil could prevent memory impairment in mice, which might be a potential way for the prevention of memory dysfunctions.

## 1. Introduction

Alzheimer’s disease (AD) is one of the most common causes of dementia with a progressive and fatal neurodegenerative disorder in the elderly [1]. At present, one pathogenesis of AD is the cholinergic hypothesis that the cholinergic system dysfunction is closely related to AD [2], which has features of aberrant oxidative [3,4], the loss of cholinergic neurons and the resultant decrease in cholinergic neurotransmission [5]. Until now, Cholinesterase inhibitors (ChEIs) have been the main approach to symptomatic treatment [6] and could help to enhance cholinergic neurotransmission by inhibiting excessive hydrolysis of acetylcholine (ACh) by acetylcholinesterase (AChE). However, these drugs’ therapeutic results are usually modest and the duration of effect and long-term safety are unknown [7]. Thus, it is necessary to find an alternative way to prevent or cure dementia.

One promising avenue to curtail the age-related cognitive and behavioral decline is dietary intervention [8]. Recently, potential benefits of nutritional supplementation with polyunsaturated fatty acid-rich plant foods have been suggested in neurodegenerative and inflammatory diseases including AD [9,10]. Vegetable oils such as olive oil, corn oil, and perilla oil have the neuroprotective effect on oxidative stress in neurons that is associated with AD [11,12]. Walnut oil is a high-valued oil product extracted from the walnut kernel. It has been shown that walnut oil is mainly composed of five fatty acids (palmitic, stearic, oleic, linoleic and linolenic acids). In summary, walnut oil provides a high level of linoleic acid (approximately 46.9%–68.6%), followed by oleic acid (10.0%–25.1%) and linolenic acid (6.9%–17.6%). In particular, polyunsaturated fatty acids are the main group of fatty acids in walnut oil (62.9%–79.1%) [13]. In addition to this, walnut oil also includes tocopherols (441.03–490.32 mg/kg), phytosterols (1014.49–1211.40 mg/kg), squalene (4.41–5.21 mg/kg), and polyphenols (44.78–64.61 mg GAE/kg) and provide good antioxidant capacities [14]. Recent research found that a diet with walnut oil may help to reduce the risk of developing age-related diseases induced by D-galactose [15]. This means that walnut oil might be used as a promising product for cognitive prevention in AD.

Scopolamine is a muscarinic receptor antagonist which can block the cholinergic function of the central nervous system [16]. Scopolamine can produce similar memory deficits seen in the elderly and significantly increase AChE activity and decrease the choline acetyltransferase (ChAT), ACh level and the activities of antioxidant enzymes in the brain of memory-impaired mice [17]. Furthermore, scopolamine could induce oxidative stress in the brain [18,19,20]. Thus, scopolamine is usually used as a standard drug for inducing cognitive deficits in healthy animals and produces an AD model for testing cognition-enhancing properties of new drugs [21].

However, few studies have involved the effects of walnut oil on cholinergic system damage. This study aimed to investigate whether walnut oil could prevent cognitive impairment and memory deficits induced by scopolamine. The Morris Water Maze (MWM) test was used to access the spatial memory and brain tissue-related indicators were detected to explore the possible mechanism. Furthermore, histological examination was performed to reflect the changes in neuronal cells.

## 2. Results

### 2.1. Effect of Walnut Oil on Spatial Memory in the Water Morris Maze

The performance of all groups in MWM training tasks is shown in Figure 1A. The behavior of mice in all groups improved during successive training days in the form of shortened escape latency. The SCOP group took a longer time to find the platform than the CON group from the second day to the fourth day of the training period, showing that SCOP treatment could cause significant cognitive dysfunction in mice. Throughout the whole training period, there was no significant difference in the mean latency between the WO group and CON group (*p* > 0.05). Meanwhile, walnut oil and piracetam treatment significantly decreased the escape latency prolonged by SCOP on the third (24.38 ± 3.68, 23.17 ± 3.42, respectively) and fourth (22.73 ± 3.51, 20.72 ± 4.70, respectively) days of the training days compared to the SCOP group (36.41 ± 4.56, 35.80 ± 3.52, respectively) (*p* = 0.027, *p* = 0.028, *p* = 0.015, *p* = 0.012).

As shown in Figure 1B, the total swimming distance of each group was decreased throughout the training days. Comparing with the CON group, the SCOP-treated mice swam significantly longer distance from the second day onwards. In addition, the swimming distance of the WO group (680.91 ± 82.23) reduced obviously on the fourth day compared with the SCOP group (913.45 ± 58.61) (*p* = 0.005).

On the fifth day, comparing to the CON group, SCOP-treatment decreased the number of target crossings to 0.69 ± 0.21 (*p* = 0.012) (Figure 1C), reduced the time spent in the garget quadrant to 11.49 ± 0.61 (*p* = 0.018) (Figure 1D), and dropped the percentage of target quadrant distance and time to 22.32% ± 1.75% (Figure 1E) and 22.01% ± 1.18% (Figure 1F) (*p* = 0.002, *p* = 0.011) respectively. However, walnut oil treatment led to a significant increase in above indexes versus the SCOP group (1.43 ± 0.30, 15.36 ± 1.46, 28.27% ± 1.48%, 30.60% ± 1.74%, respectively) (*p =* 0.035, *p =* 0.044, *p =* 0.033, *p =* 0.016). Figure 1G depicts the represent tracks of mice on the fifth day of the MWM test. Comparing to the CON group, the SCOP-treated mice took a longer time and complex swimming paths to find the platform. WO group mice performed better tracks than the SCOP group with the shortened distance and uncomplicated tracks.

In the MWM test, the WO group and PIRA group had the same trend. These results indicated that walnut oil could improve the spatial learning and memory functions of mice with memory deficits induced by SCOP.

### 2.2. Effects of Walnut Oil on Brain AChE and ChAT Activities

The effects of walnut oil on brain AChE and ChAT activities are shown in Figure 2. The activity of AChE was significantly up-regulated by SCOP (1.69 ± 0.13) (*p =* 0.003) compared to the CON group (1.17 ± 0.14). When the SCOP-treated mice were treated with walnut oil, the activity of AChE (1.26 ± 0.12) in the brain tissue was remarkably inhibited in comparison with the SCOP group (*p =* 0.013). Further, treatment with piracetam also significantly suppressed the increase of AChE activity (1.25 ± 0.08). Meanwhile, the activity of ChAT was found to be lower in the SCOP group (99.53 ± 3.89) than that in the CON group (173.76 ± 12.31) (*p =* 0.007). In addition, walnut oil (129.75 ± 6.76) obviously reversed the decrease caused by SCOP (*p =* 0.024). However, there was no significant difference between the WO group and the CON group, though the walnut oil decreased the activity of AChE (1.12 ± 0.12) and improved the activity of ChAT (183.31 ± 20.70). Notably, no obvious difference was found between the WO group and PIRA group. These results indicated that the administration of walnut oil suppressed the increase of AChE activity and the reduction of ChAT activity caused by scopolamine administration.

### 2.3. Effect of Walnut Oil on Oxidative Stress

Figure 3A,B show the changes in T-SOD activity and GSH content in brain tissues. In the SCOP group, the results showed that the injection of scopolamine significantly decreased T-SOD activity (55.17 ± 3.31) and GSH content (68.42 ± 4.57) compared to the CON group (100.15 ± 9.07 and 115.11 ± 19.85, respectively)(*p =* 0.001, *p* = 0.028). The T-SOD activity and GSH content in these tissues were significantly higher in the WO group (93.30 ± 5.50, 110.45 ± 17.70, respectively) and the PIRA group (95.56 ± 9.01, 112.32 ± 5.08, respectively) than the SCOP group (*p* = 0.006; *p* = 0.047; *p* = 0.004; *p* = 0.038). On the other hand, the changes in MDA levels in the brain tissues are shown in Figure 3C. The level of MDA got a significant increase of up to 19.19 ± 1.71 nmol/mg in the SCOP group from 10.26 ± 0.72 nmol/mg in the CON group. Yet, the increase of the MDA level caused by SCOP was significantly diminished by walnut oil (13.79 ± 0.96) and piracetam (13.24 ± 1.04). There was no significant difference among the WO group and the CON group and PIRA group in the above brain tissues biochemical parameters.

### 2.4. Effect of Walnut Oil on Hippocampal CA1 and CA3 Regions

Nissl staining of pyramidal cells in the hippocampal CA1 and CA3 areas of mice indicated an extensive damage with large number of dark-stained shrunken neurons cells in the SCOP group whereas the CON group was normal with well outlined and closely arranged soma. Such degenerate neurons were significantly reduced in CA1 and CA3 regions in memory impaired mice administrated with walnut oil and piracetam (Figure 4). The number of live neurons of the hippocampal CA1 and CA3 regions were counted in all groups (Figure 5A,B). Based on the results, scopolamine markedly decreased the numbers of live cells in the hippocampal CA1 (*p* = 0.000) and CA3 (*p* = 0.000) regions as compared with the CON group. In addition, when comparing to the SCOP group, a significant increase of live neurons of the hippocampal CA1 (*p* = 0.002) and CA3 (*p* = 0.006) regions of WO group was observed. Moreover, walnut oil treatment had no effect on the cell morphology and live neurons compared to the mice in the CON group and the PIRA group.

## 3. Discussion

In this study, the results revealed that SCOP treatment could induce spatial learning and memory functions impairments, cholinergic system dysfunction and oxidative stress in the brain tissue of mice. However, walnut oil could make these changes caused by SCOP better via enhancing cholinergic function and inhibiting oxidative damage in the brain of the SOCP-treated mice. That is to say, walnut oil has a protective effect against scopolamine modulation as piracetam.

To our knowledge, memory was assessed by WMW test that is generally used to evaluate spatial learning ability and long-term spatial memory. Behavioral data obtained in our study indicated that SCOP-treatment delayed the escape latency, increased the swimming distance during the trial sessions, and decreased the number of crossing over the platform, shortened the time in the target quadrant, dropped the percentage of target quadrant distance and time in the probe test. These results are in accordance with some previous studies [22,23].Walnut oil showed inhibitory effects on SCOP-induced memory impairment in the MWM test, that is, mice treated with walnut oil showed shorter escape latency and swimming distance, improved the platform-crossing numbers, the swimming time within the target quadrant, and the percentage of target quadrant distance and time, suggesting that walnut oil could ameliorate SCOP-induced behavioral impairment.

The central cholinergic system is closely related to learning and memory functions [24] and ACh widely scattered in the nervous system is one of the important neurotransmitters and plays a critical role in modulating cognitive performances and learning and memory processes [11,25]. Cholinergic transmission mainly releases ACh from vesicles and transmits ACh to the synaptic cleft where ACh is hydrolyzed by AChE to acetic acid and choline [26]. However, overmuch AChE activity may lead to ACh deficiency and cognitive dysfunction [27,28]. At the same time, ChAT can help to catalyze the biosynthesis of acetylcholine [29]. Thus changes in AChE and ChAT activities are often used as cholinergic biomarkers for diagnose AD [30]. In this study, walnut oil treatment significantly inhibited AChE activity and promoted ChAT activity in the brain of cognitive deficient mice. The result was consistent with behavioral data, indicating that walnut oil could provide neuroprotection via regulating the stability of cholinergic neuronal systems.

Oxidative stress is one of the main factors in the pathogenesis and aggravation of neurodegenerative disorders like AD [31]. Superoxide dismutase (SOD), glutathione (GSH) and malondialdehyde (MDA) act as the three major biomarkers of oxidative stress. SOD and GSH are important antioxidants inhibiting the production of free radicals [32,33], while MDA is the product of biofilm lipid peroxidation which is generally considered to be a marker of radical generation [34]. Intraperitoneal injection of scopolamine can cause oxidative damage in the brain via triggering the production of ROS. In our findings, SOD activity and GSH content were decreased and the content of MDA was increased in the brain of scopolamine-treated mice. The brain is highly enriched with polyunsaturated fatty acids, especially arachidonic acid and docosahexaenoic acid. The two fatty acids could help modulate synaptic plasticity, inhibit neuroinflammation, prevent oxidative stress and protect neurons [35,36]. Therefore, appropriate supplementation of EPA or DHA can improve cognitive impairment in elderly individuals [37]. Long-term dietary intake of α-linolenic acid (ALA) can improve the ability of learning and memory by regulating fatty acid content in the brain of patients with mild memory or cognitive impairment [38]. However, excessive proportions of n-6/n-3 fatty acids in the diet can cause mental retardation and brain damage like in Alzheimer’s disease [39], and the decreased ratio of the n-6/n-3 fatty acid can help to regulate brain innate immune system activity, resist LPS-induced proinflammatory cytokine production and subsequent spatial memory changes [40]. Therefore, it is especially important to maintain a balanced ratio of n-6/n-3 fatty acids for brain development. An effective ratio of ALA and linoleic acid (LA) in walnut oil may be a contributing factor in improving the antioxidant capacity of the brain in memory-impaired mice [41]. Moreover, high levels of tocopherols, phytosterols, squalene and polyphenols in walnut oil play an important role in the good antioxidant capacity [14]. The present findings indicated that walnut oil increased SOD activity and GSH content and decreased the level of MDA markedly. At the same time, these results are be consistent with the finding that walnut oil treatment can protect against deficits in neurons of hippocampal CA1 and CA3 regions induced by scopolamine, which implied that the neuroprotective action of walnut oil is in relation to its antioxidant activity.

## 4. Materials and Methods

### 4.1. Animals

A total of 40 eight-week-old male specific pathogen free Kunming mice (20–24 g) obtained from the Xi’an Jiaotong University Experimental Animal Center were used for the study. Mice were housed in standard conditions of 23 ± 1 °C and 55% humidity with 12:12 h light/dark cycle. The mice were given laboratory food and purified water ad libitum. All animal experiments were carried out in strict compliance with “Guide of the care and use of laboratory animals’’ (National Research Council, 1996).

### 4.2. Treatment

In the experiment, mice were randomly divided into four groups (10 mice each): Control (CON), Scopolamine (SCOP), Piracetam (PIRA) and Walnut Oil (WO). The mice in the PIRA group, a positive control group, were orally administered with 800 mg/kg piracetam daily and the WO group were orally received 10 mL/kg walnut oil daily for eight consecutive weeks [42]. During the experimental period, the CON and SCOP groups were given purified water by gavage.

The Morris water maze (MWM) test started on day 51. After the oral administration in all groups, the mice in CON group were received 0.9% saline by intraperitoneal injection and the other three groups (SCOP, PIRA and WO) were received 3 mg/kg scopolamine in the same way. The MWM test began after 30 min.

### 4.3. Morris Water Maze

The Morris Water Maze is one of the most widely used spatial learning and memory testing programs described by Morris [43] and was modified in the present experiment.

The water maze consists of a circular pool which is 120 cm diameter and 50 cm height and a camera placed over the labyrinth to record the movement track of the mouse. The water temperature in pool was maintained at 21~23 °C. The pool is divided into four equal quadrants and a platform is placed in the middle of the fourth quadrant. Before the experiment, the mice were placed in the pool to swim freely for 1 min to adapt to the water temperature and to be familiar with the environment. During the training period, the visible platform was placed 1 cm above the water surface on the first day, and the mice were requested to stand on the platform for 30 s and then released into water facing the tank wall from three different quadrants and permitted to swim within 60 s. When reaching the platform successfully, they had to stand on the platform for 30 s. If they could not, the experimenter needed to guide them to the platform and allow them to stay for 30 s. During the following four consecutive days, the platform was placed 1 cm below the surface of the water and melanin was added to the water. Recording the effective time that the mice could reach the platform and stay for 3 s as the escape latency. For mice that failed to find the platform in 60 s, the escape latency was recorded as 60 s, and the mice was guided to the platform for 30 s. After the training sessions, the platform was removed and other condition remained the same. Dates of each day were recorded for analysis.

### 4.4. Biochemical Determination

After the MWM test, these mice were sacrificed by decapitation and their brain samples were removed. Part of these brains were snap-frozen in liquid nitrogen before storage at −80 °C for biochemical determination The SOD activity, GSH content, MDA content, AChE and ChAT in brain tissue were determined using commercial kits by the methods of hydroxylamine, Spectrophotometry, TBA (Thiobarbituric acid), Spectrophotometry, ultraviolet colorimetry, respectively.

### 4.5. Morphological Examination

These mice were sacrificed and another part of the brain samples were fixed in 4% paraformaldehyde preparation overnight. Then, these samples were taken out, embedded in paraffin, cut into 3 μm thick sections, and stained with cresyl purple acetate. The hippocampal CA1 and CA3 regions of the mouse brain tissue were examined under a bright field illumination using the DM5000 B fluorescence microscope, and viable cells were counted at 400-fold magnification. In this study, intact cells were considered as the viable cells and were used for cell counting.

### 4.6. Statistical Analysis

Data concerning the WMW test and biochemical parameters were expressed as mean ± SEM. Results were analyzed by using one-way analysis of variance (ANOVA) followed by Tukey’s multiple comparison test using SPSS 19.0 software (IBM, Chicago, USA) for windows to look for intergroup differences. During the MWM training task, group differences in the escape latency were analyzed by two-way ANOVA. Statistical significance was considered if the *p* value was less than 0.05.

## 5. Conclusions

In this study, walnut oil improved the performance of mice in Morris Water Maze tests by shortening the escape latency and total swimming distance and increasing the number of target crossings, the time spent in the garget quadrant, the percentage of target quadrant distance and the time. It also regulated the activities of cholinergic biomarkers and inhibited the oxidative stress and ameliorated the cell morphology. Hence, walnut oil could be a promising supplement in the prevention of memory dysfunction. However, the components of walnut oil that have protective effects on learning and memory and its mechanism of action need further study.

## Figures and Tables

**Figure 1 molecules-25-01630-f001:**
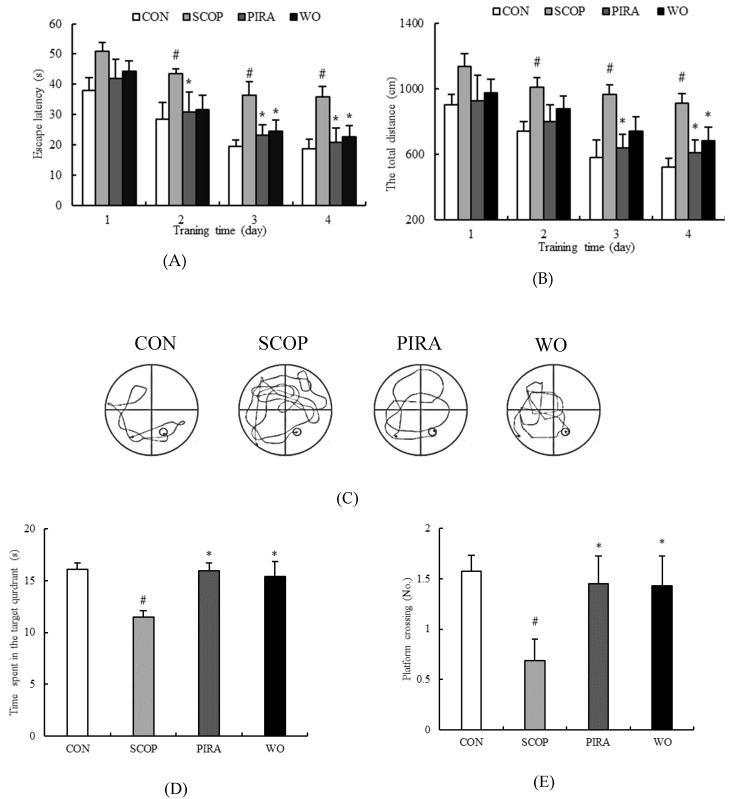
Effects of walnut oil on escape latency (**A**) and total swimming distance (**B**) from day1 to day 4, and number of platform crossing (**C**), time spent in the target quadrant (**D**), percentage of the target quadrant distance (**E**), percentage of the target quadrant time (**F**) and represent tracks of mice (**G**) on day 5 during the Morris water maze test. Data are shown as mean ± SEM. # *p* < 0.05 versus the control group. **p* < 0.05 versus the SCOP-treated group.

**Figure 2 molecules-25-01630-f002:**
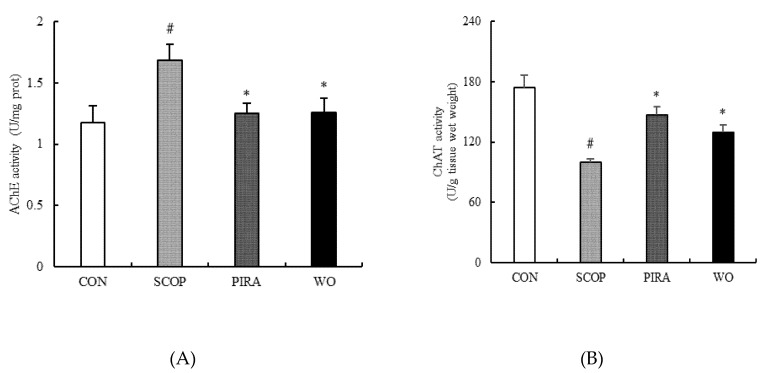
Effects of walnut oil on the activity of AChE (**A**) and ChAT (**B**) in brains of mice. Data are shown as mean ± SEM. # *p* < 0.05 versus the control group. **p* < 0.05 versus the SCOP-treated group.

**Figure 3 molecules-25-01630-f003:**
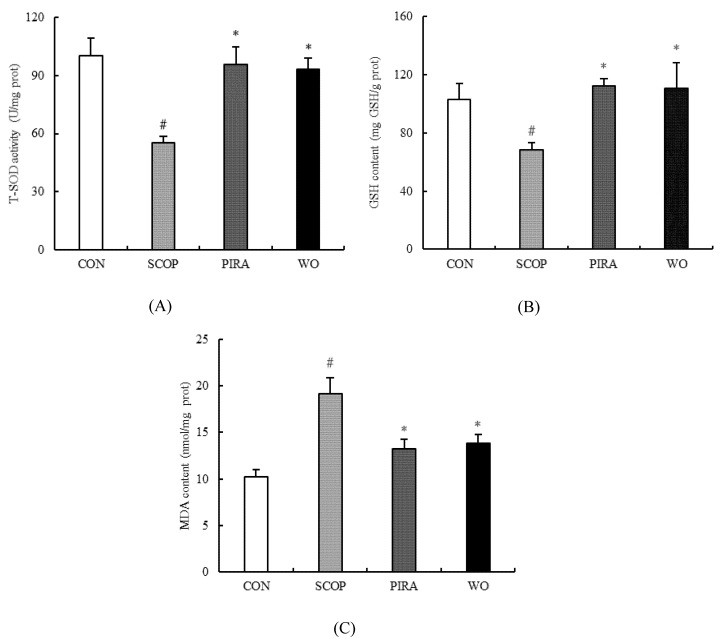
Effects of walnut oil on the activity of T-SOD (**A**), the contents of GSH (**B**) and MDA (**C**) in brains of mice. Data are shown as mean ± SEM. #*p* <0.05 versus the control group. **p* <0.05 versus the SCOP-treated group.

**Figure 4 molecules-25-01630-f004:**
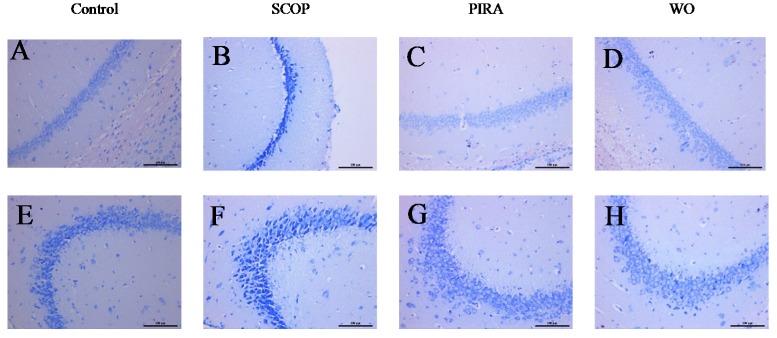
Effects of walnut oil on the result of Nissl staining of hippocampal CA1 (**A**–**D**) and CA3 (**E**–**H**) regions of brains of mice. Histological sections of the brain tissue showing neurological lesions (**A**–**H**). Scale bar is 100 µm.

**Figure 5 molecules-25-01630-f005:**
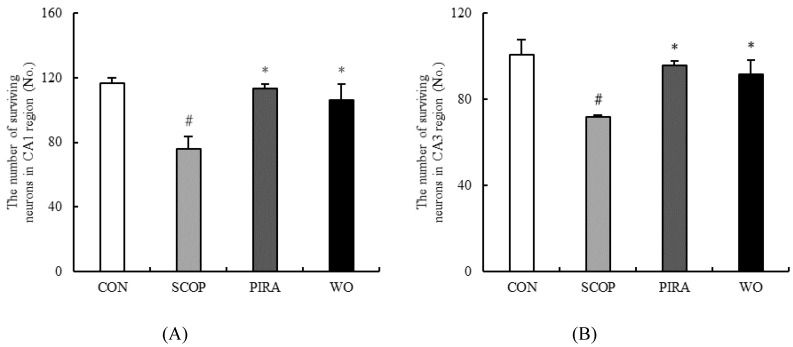
Effect of walnut on the number of surviving neurons in hippocampal CA1 (**A**) and CA3 (**B**) regions. Data are shown as mean ± SEM. # *p* <0.05 versus the control group. **p* <0.05 versus the SCOP-treated group.

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
