# Peer review of "Walnut Oil Prevents Scopolamine-Induced Memory Dysfunction in a Mouse Model"

_molecules, 2020, doi:10.3390/molecules25071630_

Round 1

Reviewer 1 Report

In this study, Jianqiao Liao et al. showed that walnut oil can prevent scopolamine-induced memory dysfunction in mice model. The overall manuscript is well organized and written with little typographic errors (for example, line 79: "oi" you mean oil? forgotten spaces - line 120: "respectively)than", ...). Please, check it.

My questions and comments:

- Please, check and unite the color and style of all figures.

- In text (line 117 and 243) you wrote about GSH activity, but fig.3B shows concentration of GSH. What is right?

- In same figure 3 (line 126), you wrote: "concentration of MDA (B) and GSH (C)", but figures show GSH (B) and MDA (C). Please, change the designation.

- In discussion part (line 149 and 198) you wrote: "walnut oil did not have any influence on the learning and memory functions on mice" and "The present findings showed that walnut oil increased the activities of SOD and GSH and decreased the level of MDA markedly" Did you measured effect of walnut oil alone? I understand that walnut oil is only in combination with scopolamine.

--If yes, please add the results to the manuscript or supplementary results.

--If not, this statement is just speculation without results and I recommend rewriting these sentences in the sense that walnut oil has a protective effect agains scopolamine modulation.

In my opinion, in line 213 and 215 are the same information about concentration of used compounds, so I recommend to keep only on one sentence. 

Finally, in next study it would be good to focus on specific or abundant walnut oil fatty acid not only crude sample.

Reviewer 2 Report

The authors have used a scopolamine model of memory impairment to show that walnut oil treatment wipes out the memory impairment resulting from scopolamine treatment to Kunming mice. The following are my concerns:

1) The language and style of the manuscript needs a thorough re-work.

2) There are several publications on the beneficial effects of walnut on oxidative stress as well as memory impairment (e.g. Effect of walnut protein hydrolysate on scopolamine-induced learning and memory deficits in mice. Li W, Zhao T, Zhang J, Xu J, Sun-Waterhouse D, Zhao M, Su G. J Food Sci Technol. 2017 Sep;54(10):3102-3110. doi: 10.1007/s13197-017-2746-x. Epub 2017 Jul 5). Thus there is novelty in this manuscript. If the authors want to stress that the walnut oil is better  than other forms of dietary walnut supplement, then the study design does not address that.

3) Other authors have tested walnut in proven models of AD using the transgenic mouse model (e.g. Chauhan et al. Dietary Supplementation of Walnuts Improves Memory Deficits and Learning Skills in Transgenic Mouse Model of Alzheimer's Disease. Journal of Alzheimer's Disease, Volume 42, Number 4 / 2014 DOI: 10.3233/JAD-140675)

4) What is the justification for the doses and treatment length for Walnut oil, scopolamine, piracetam? Other than the training results, no other result correlates the treatment day with the result. For example the Morris water maze testing shows training days up to 5 while we do not know when the animals were sacrificed to collect brains either for histology or for enzyme analyses. If the behavior test was completed in five days, then why provide treatment for 8 weeks? Or are the behavior tests were performed on the seventh week? While the authors describe piracetam and walnut oil were administered for eight weeks, they fail to describe the duration for scopolamine treatment.

5) Scopolamine treatment has been shown to differ in its effects on different breeds of mice (see  https://www.ncbi.nlm.nih.gov/pmc/articles/PMC6333609/

How does the effects in Kunming mice compare with other breeds of mice commonly used in research?

6) In almost all results the Piracetam group performs very similar to the control. Why did the authors choose piracetam as positive control, is it due to its influence on muscarinic receptors or AMPA receptors? Do the results shown in this manuscript demonstrate almost all of the effects of scopolamine on memory impairment due to its action on muscarinic receptors?  

7) If scopolamine treatment blocks muscarinic receptors (antagonist of muscarinic cholinergic receptors) to bring about learning and memory deficits, what is the use of reduced AChE and increased ChAT if the muscarinic receptors are still blocked by scopolamine? With continued treatment with scopolamine what happens to muscarinic receptor population expression?

8) Why authors measured AChE activity as U/MG protein while they measured ChAT activity as U/MG wet weight of tissue?

9) Fig A and B show that both the latency and distance of swimming is getting gradually reduced with each increasing day of training in SCOP group as in all other groups although from day 1 SCOP group showed increased latency and distance compared to other groups. Will like the authors to graph % of reduction from each group’s performance keeping day 1 as 100% to see if there is any memory impairment in SCOP group if that group also shows signs of learning over time 

10) Histology pictures (Nissl stain) do not show any detail. Authors should have stained specifically for cell death

Reviewer 3 Report

In this study the authors provide new findings about the possible pathway involved in the neuroprotective effects of walnut oil. The reported data show that walnut oil is able to prevent alteration of cholinergic system also throughout its antioxidant activity thus preventing memory impairment induced by scopolamine in mice. The study is interesting, but several issues have to be clarified.

Major points:

Line 46: The ratio of n−6/n−3 PUFAs in walnut oil must be indicated in the text. It has been characterized the chemical composition of walnut oil? Describe it and report the appropriate reference. The authors attribute the protective effects of walnut oil on cholinergic system and oxidative stress only to the PUFAs content. However other important phytonutrients likely contributing to neuronal protection are contained in walnut oil, but these components are not mentioned in the text. A deeper discussion about literature data reporting the neuroprotective effects of different walnut oil components both in the introduction and discussion sections is necessary to make the manuscript complete. More details regarding the animal treatment are needed: specify the methods and times used to administer the different compounds, for example orally or i.p. , and daily or weekly. Why did the authors choose the concentration of 10 mL/kg walnut oil for WO group? What about possible effects of higher or lower concentrations in mice? In the first sentence of the discussion the authors state that walnut oil did not have any influence on learning and memory functions in mice compared to the control group. To sustain this hypothesis it would be necessary to treat a group of mice only with walnut oil, but in the present study this set of experiments has not been performed, only the WO effects against scopolamine have been evaluated. The sentence must be removed or changed accordingly, or new experiments need to be added. The comparison between PIRA and WO effects should be described in all results paragraphs and discussed. In the paragraph “Effect of walnut oil on oxidative stress” the terms activity, levels and amount must be appropriately used. The authors did not measured gsh activity or T-SOD levels.

Minor points:

Throughout the text the authors define the WO group as WOP group, it must be corrected. Line 144: I think that scale bar of 100 mm is another mistake. In the results paragraph it is appropriate to report the exact p-values rather than p<0.05. The p-values reported in lines 137 and 139 must be removed: put the exact p-values near CA1 or CA3 respectively. In Figure 5 the number of viable cells is shown, but in the methods section it is not detailed the size of the area considered for cell counting in the different hippocampal CA1 and CA3 regions.

Round 2

Reviewer 1 Report

The authors have made a great effort to improve the quality of the manuscript. All of the concerns in the previous comments have been properly addressed.

Author Response

Point 1: The authors have made a great effort to improve the quality of the manuscript. All of the concerns in the previous comments have been properly addressed.

ʉ۬

Response 1: Thanks very much for the reviewer’s acceptance of the manuscript.

Reviewer 3 Report

The manuscript has been improved, however some issues have not been addressed.

The English language of the manuscript needs further improvement. The composition of walnut oil must be described in the introduction, not only reporting the main fatty acids listed in response 1, but also referring to other types of molecules (Pan Gao, Ruijie Liu, Qingzhe Jin, Xingguo Wang, Comparative Study of Chemical Compositions and Antioxidant Capacities of Oils Obtained From Two Species of Walnut: Juglans Regia and Juglans Sigillata. Food Chem 2019, 279, 279-287 ). A deeper discussion about literature data reporting the protective effects of walnut oil components in the discussion sections is necessary to make the manuscript complete. The observations reported in response 3 and 9 must be added and discussed in the text. Although in several researches the significant difference between groups is expressed as a p-value less than 0.05, the guidelines for reporting statistics of many journals recommend to express the actual P value (for example P=.04) rather than expressing a statement of inequality (P<.05), unless P<.001, thus providing a clearer interpretation of results. The SPSS software used by the authors to analyse statistical differences provides the actual pvalue.

Author Response

Point 1: The English language of the manuscript needs further improvement.

Response 1: The English language of the manuscript has been improved in the revised manuscript and has been checked by a native English speaker.

Point 2: The composition of walnut oil must be described in the introduction, not only reporting the main fatty acids listed in response 1, but also referring to other types of molecules (Pan Gao, Ruijie Liu, Qingzhe Jin, Xingguo Wang, Comparative Study of Chemical Compositions and Antioxidant Capacities of Oils Obtained From Two Species of Walnut: Juglans Regia and Juglans Sigillata. Food Chem 2019, 279, 279-287 ). A deeper discussion about literature data reporting the protective effects of walnut oil components in the discussion sections is necessary to make the manuscript complete.

Response 2: Thanks for the reviewer’s valuable advice. The composition of walnut oil has been supplemented in the introduction in the revised manuscript. Meanwhile, literature data reporting the protective effects of walnut oil components has also been added in the discussion sections. Thanks again for the reviewer’s suggestion.

Point 3: The observations reported in response 3 and 9 must be added and discussed in the text.

Response 3: Thanks for the reviewer’s suggestion. The previous observation has been added in the revised manuscript.

Point 4: Although in several researches the significant difference between groups is expressed as a p-value less than 0.05, the guidelines for reporting statistics of many journals recommend to express the actual P value (for example P=.04) rather than expressing a statement of inequality (P<.05), unless P<.001, thus providing a clearer interpretation of results. The SPSS software used by the authors to analyse statistical differences provides the actual pvalue.

Response 4: The actual P value has been expressed in the revised manuscript according to the reviewer’s suggestion.